# How to Train Your Super-Net: An Analysis of Training Heuristics in Weight-Sharing NAS

## Abstract

Weight sharing promises to make neural architecture search (NAS) tractable even on commodity hardware. Existing methods in this space rely on a diverse set of heuristics to design and train the shared-weight backbone network, a.k.a. the super-net. Since heuristics substantially vary across different methods and have not been carefully studied, it is unclear to which extent they impact super-net training and hence the weight-sharing NAS algorithms. In this paper, we disentangle super-net training from the search algorithm, isolate 14 frequently-used training heuristics, and evaluate them over three benchmark search spaces. Our analysis uncovers that several commonly-used heuristics negatively impact the correlation between super-net and stand-alone performance, whereas simple, but often overlooked factors, such as proper hyper-parameter settings, are key to achieve strong performance. Equipped with this knowledge, we show that simple random search achieves competitive performance to complex state-of-the-art NAS algorithms when the super-net is properly trained.

## 1 Introduction

Neural architecture search (NAS) has received growing attention in the past few years, yielding state-of-the-art performance on several machine learning tasks (Liu et al., 2019a; Wu et al., 2019; Chen et al., 2019b; Ryoo et al., 2020). One of the milestones that led to the popularity of NAS is weight sharing (Pham et al., 2018; Liu et al., 2019b), which, by allowing all possible network architectures to share the same parameters, has reduced the computational requirements from thousands of GPU hours to just a few. Figure 1 shows the two phases that are common to weight-sharing NAS (WS-NAS) algorithms: the search phase, including the design of the search space and the search algorithm; and the evaluation phase, which encompasses the final training protocol on the proxy task [1].

While most works focus on developing a good sampling algorithm (Cai et al., 2019; Xie et al., 2019) or improving existing ones (Zela et al., 2020a; Nayman et al., 2019; Li et al., 2020), they tend to overlook or gloss over important factors related to the design and training of the shared-weight backbone network, i.e. the super-net. For example, the literature encompasses significant variations of learning hyper-parameter settings, batch normalization and dropout usage, capacities for the initial layers of the network, and depth of the super-net. Furthermore, some of these heuristics are directly transferred from standalone network training to super-net training without carefully studying their impact in this drastically different scenario. For example, the fundamental assumption of batch normalization that the input data follows a slowly changing distribution whose statistics can be tracked during training is violated in WS-NAS, but nonetheless typically assumed to hold.

In this paper, we revisit and systematically evaluate commonly-used super-net design and training heuristics and uncover the strong influence of certain factors on the success of super-net training. To this end, we leverage three benchmark search spaces, NASBench-101 (Ying et al., 2019), NASBench-201 (Dong & Yang, 2020), and DARTS-NDS (Radosavovic et al., 2019), for which the ground-truth stand-alone performance of a large number of architectures is available. We report the results of our experiments according to two sets of metrics: i) metrics that directly measure the quality of the super-net, such as the widely-adopted super-net accuracy [2] and a modified Kendall-Tau correlation between the searched architectures and their ground-truth performance, which we refer to as *sparse*

---

[1] Proxy task refers to the tasks that neural architecture search aims to optimize on.
[2] The mean accuracy over a small set of randomly sampled architectures during super-net training.

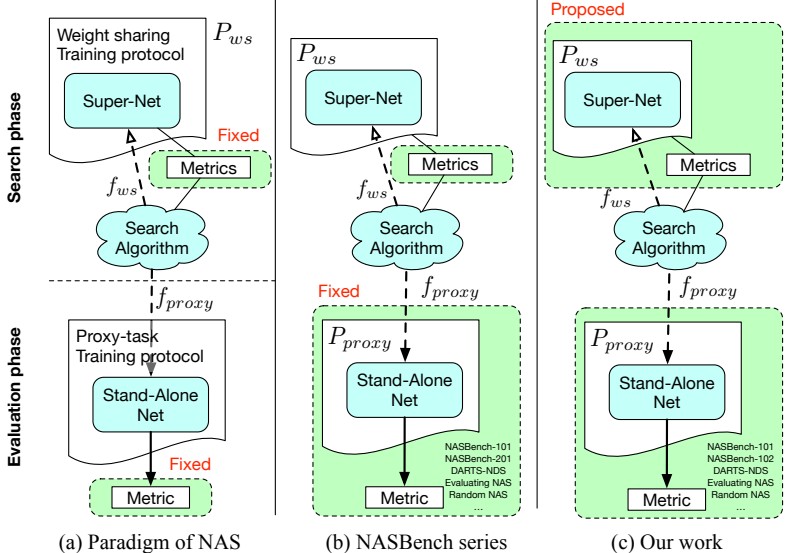

Figure 1: **WS-NAS benchmarking.** Green blocks indicate which aspects of NAS are benchmarked in different works. A search algorithm usually consists of a search space that encompass many architectures, and a policy to select the best one. $P$ indicates a training protocol, and $f$ a mapping function from the search space to a neural network. **(a)** Early works fixed and compared the metrics on the proxy task, which doesn't allow for a holistic comparison between algorithms. **(b)** The NASBench benchmark series partially alleviates the problem by sharing the stand-alone training protocol and search space across algorithms. However, the design of the weight-sharing search space and training protocol is still not controlled. **(c)** We fill this gap by benchmarking existing techniques to construct and train the shared-weight backbone. We provide a controlled evaluation across three benchmark spaces.

*Kendall-Tau*; ii) proxy metrics such as the ability to surpass random search and the stand-alone accuracy of the model found by the WS-NAS algorithm.

Via our extensive experiments (over 700 GPU days), we uncover that (i) the training behavior of a super-net drastically differs from that of a standalone network, e.g., in terms of feature statistics and loss landscape, thus allowing us to define training factor settings, e.g., for batch-normalization (BN) and learning rate, that are better suited for super-nets; (ii) while some neglected factors, such as the number of training epochs, have a strong impact on the final performance, others, believed to be important, such as path sampling, only have a marginal effect, and some commonly-used heuristics, such as the use of low-fidelity estimates, negatively impact it; (iii) the commonly-adopted super-net accuracy is unreliable to evaluate the super-net quality.

Altogether, our work is the first to systematically analyze the impact of the diverse factors of super-net design and training, and we uncover the factors that are crucial to design a super-net, as well as the non-important ones. Aggregating these findings allows us to boost the performance of simple weight-sharing random search to the point where it reaches that of complex state-of-the-art NAS algorithms across all tested search spaces. We will release our code and trained models so as to establish a solid baseline to facilitate further research.

## 2 PRELIMINARIES AND RELATED WORK

We first introduce the necessary concepts that will be used throughout the paper. As shown in Figure 1(a), weight-sharing NAS algorithms consist of three key components: a search algorithm that samples an architecture from the search space in the form of an encoding, a mapping function $f_{proxy}$ that maps the encoding into its corresponding neural network, and a training protocol for a proxy task $P_{proxy}$ for which the network is optimized.

To train the search algorithm, one needs to additionally define the mapping function $f_{ws}$ that generates the shared-weight network. Note that the mapping $f_{proxy}$ frequently differs from $f_{ws}$, since in practice the final model contains many more layers and parameters so as to yield competitive results on the proxy task. After fixing $f_{ws}$, a training protocol $P_{ws}$ is required to learn the super-net. In practice, $P_{ws}$ often hides factors that are critical for the final performance of an approach, such

as hyper-parameter settings or the use of data augmentation strategies to achieve state-of-the-art performance (Liu et al., 2019b; Chu et al., 2019; Zela et al., 2020a). Again, $P_{ws}$ may differ from $P_{proxy}$, which is used to train the architecture that has been found by the search. For example, our experiments reveal that the learning rate and the total number of epochs frequently differ due to the different training behavior of the super-net and stand-alone architectures.

Many strategies have been proposed to implement the search algorithm, such as reinforcement learning (Zoph & Le, 2017; Zoph et al., 2018), evolutionary algorithms (Real et al., 2017; Miikkulainen et al., 2019; So et al., 2019; Liu et al., 2018; Lu et al., 2018), gradient-based optimization (Liu et al., 2019b; Xu et al., 2020; Li et al., 2020), Bayesian optimization (Kandasamy et al., 2018; Jin et al., 2019; Zhou et al., 2019; Wang et al., 2020), and separate performance predictors (Liu et al., 2018; Luo et al., 2018). Until very recently, the common trend to evaluate NAS consisted of reporting the searched architecture's performance on the proxy task (Xie et al., 2019; Real et al., 2019; Ryoo et al., 2020). This, however, hardly provides real insights about the NAS algorithms themselves, because of the many components involved in them. Many factors that differ from one algorithm to another can influence the performance. In practice, the literature even commonly compares NAS methods that employ different protocols to train the final model.

Li & Talwalkar (2019) and Yu et al. (2020b) were the first to systematically compare different algorithms with the same settings for the proxy task and using several random initializations. Their surprising results revealed that many NAS algorithms produce architectures that do not significantly outperform a randomly-sampled architecture. Yang et al. (2020) highlighted the importance of the training protocol $P_{proxy}$. They showed that optimizing the training protocol can improve the final architecture performance on the proxy task by three percent on CIFAR-10. This non-trivial improvement can be achieved regardless of the chosen sampler, which provides clear evidence for the importance of unifying the protocol to build a solid foundation for comparing NAS algorithms.

In parallel to this line of research, the recent series of "NASBench" works (Ying et al., 2019; Zela et al., 2020b; Dong & Yang, 2020) proposed to benchmark NAS approaches by providing a complete, tabular characterization of a search space. This was achieved by training every realizable stand-alone architecture using a fixed protocol $P_{proxy}$. Similarly, other works proposed to provide a partial characterization by sampling and training a sufficient number of architectures in a given search space using a fixed protocol (Radosavovic et al., 2019; Zela et al., 2020a; Wang et al., 2020).

While recent advances for systematic evaluation are promising, no work has yet thoroughly studied the influence of the super-net training protocol $P_{ws}$ and the mapping function $f_{ws}$. Previous works (Zela et al., 2020a; Li & Talwalkar, 2019) performed hyper-parameter tuning to evaluate their own algorithms, and focused only on a few parameters. We fill this gap by benchmarking different choices of $P_{ws}$ and $f_{ws}$ and by proposing novel variations to improve the super-net quality.

Recent works have shown that sub-nets of super-net training can surpass some human designed models without retraining (Yu et al., 2020a; Cai et al., 2020) and that reinforcement learning can surpass the performance of random search (Bender et al., 2020). However, these findings are still only shown on MobileNet-like search spaces where we only search for the size of convolution kernels and the channel ratio for each layer. This is an effective approach to discover a compact network, but it does not change the fact that on cell-based search space super-net quality remains low.

## 3 EVALUATION METHODOLOGY

We first isolate 14 factors that need to be considered during the design and training of a super-net, and then introduce the metrics to evaluate the quality of the trained super-net. Note that these factors are agnostic to the search policy that is used after training the super-net.

### 3.1 DISENTANGLING THE SUPER-NET FROM THE SEARCH ALGORITHM

Our goal is to evaluate the influence of the super-net mapping $f_{ws}$ and weight-sharing training protocol $P_{ws}$. As shown in Figure 2, $f_{ws}$ translates an architecture encoding, which typically consists of a discrete number of choices or parameters, into a neural network. Based on a well-defined mapping, the super-net is a network in which every sub-path has a one-to-one mapping with an architecture encoding (Pham et al., 2018). Recent works (Xu et al., 2020; Li et al., 2020; Ying et al.,

| WS Mapping $f_{ws}$ | | WS Protocol $P_{ws}$ | |
|---|---|---|---|
| *implementation* | *low fidelity* | *hyperparam.* | *sampling* |
| Dynamic Channeling | # layer | batch-norm | FairNAS |
| OFA Conv | train portion | learning rate | Random-NAS |
| WSBN | batch size | epochs | Random-A |
| Dropout | # channels | weight decay | |
| Op on Node/Edge | | | |

Table 1: Summary of factors

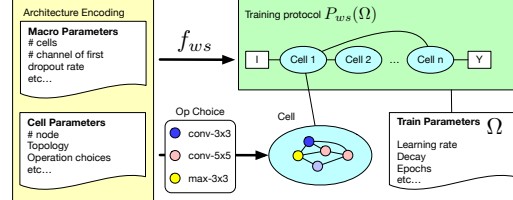

Figure 2: Constructing a super-net

2019) separate the encoding into *cell parameters*, which define the basic building blocks of a network, and *macro parameters*, which define how cells are assembled into a complete architecture.

**Weight-sharing mapping $f_{ws}$.** To make the search space manageable, all cell and macro parameters are fixed during the search, except for the topology of the cell and its possible operations. However, the exact choices for each of these fixed factors differ between algorithms and search spaces. We report the common factors in the left part of Table 1. They include various implementation choices, e.g., the use of convolutions with a dynamic number of channels (Dynamic Channeling), super-convolutional layers that support dynamic kernel sizes (OFA Kernel) (Cai et al., 2020), weight-sharing batch-normalization (WSBN) that tracks independent running statistics and affine parameters for different incoming edges (Luo et al., 2018), and path and global dropout (Pham et al., 2018; Luo et al., 2018; Liu et al., 2019b). They also include the use of low-fidelity estimates (Elsken et al., 2019) to reduce the complexity of super-net training, e.g., by reducing the number of layers (Liu et al., 2019b) and channels (Yang et al., 2020; Chen et al., 2019a), the portion of the training set used for super-net training (Liu et al., 2019b), or the batch size (Liu et al., 2019b; Pham et al., 2018; Yang et al., 2020).

**Weight-sharing protocol $P_{ws}$.** Given a mapping $f_{ws}$, different training protocols $P_{ws}$ can be employed to train the super-net. Protocols can differ in the training hyper-parameters and the sampling strategies they rely on. We will evaluate the different hyper-parameter choices listed in the right part of Table 1. This includes the initial learning rate, the hyper-parameters of batch normalization, the total number of training epochs, and the amount of weight decay.

We randomly sample one path to train the super-net (Guo et al., 2019), which is also known as single-path one-shot (SPOS) or Random-NAS (Li & Talwalkar, 2019). The reason for this choice is that Random-NAS is equivalent to the initial state of many search algorithms (Liu et al., 2019b; Pham et al., 2018; Luo et al., 2018), some of which even freeze the sampler training so as to use random sampling to warm-up the super-net (Xu et al., 2020; Dong & Yang, 2019b). Note that we also evaluated two variants of Random-NAS, but found their improvement to be only marginal. Please see Appendix C.2 for more detail.

In our experiments, for the sake of reproducibility, we ensure that $P_{ws}$ and $P_{proxy}$, as well as $f_{ws}$ and $f_{proxy}$, are as close to each other as possible. For the hyper-parameters of $P_{ws}$, we cross-validate each factor following the order in Table 1, and after each validation, use the value that yields the best performance in $P_{proxy}$. For all other factors, we change one factor at a time.

**Search spaces.** We use three commonly-used search spaces, for which a large number of stand-alone architectures have been trained and evaluated on CIFAR-10 (Krizhevsky et al., 2009) to obtain their ground-truth performance. In particular, we use NASBench-101 (Ying et al., 2019), which consists of $423,624$ architectures and is compatible with weight-sharing NAS (Yu et al., 2020b; Zela et al., 2020b); NASBench-201 (Dong & Yang, 2020), which contains more operations than NASBench-101 but fewer nodes; and DARTS-NDS (Radosavovic et al., 2019) that contains over $10^{13}$ architectures of which a subset of 5000 models was sampled and trained in a stand-alone fashion. See Appendix A.2 for a detailed discussion.

### 3.2 SPARSE KENDALL-TAU - A NOVEL SUPER-NET EVALUATION METRIC

We define a novel super-net metric, which we name *sparse Kendall-Tau*. It is inspired by the Kendall-Tau metric used by Yu et al. (2020b) to measure the discrepancy between the ordering of stand-alone architectures and the ordering that is implied by the trained super-net. An ideal super-net should yield the same ordering of architectures as the stand-alone one and thus would lead to a high Kendall-Tau. However, Kendall-Tau is not robust to negligible performance differences between architectures (c.f. Figure 3). To robustify this metric, we share the rank between two architectures if their stand-alone

accuracies differ by less than a threshold (0.1% here). Since the resulting ranks are sparse, we call this metric *sparse Kendall-Tau* (s-KdT). Note that we also compare Kendall-Tau and Spearman correlation in Appendix A.3, and provide implementation details in Appendix A.4.

Although, sparse Kendall-Tau captures the super-net quality well, it may fail in extreme cases, such as when the top-performing architectures are ranked perfectly while poor ones are ordered randomly. To account for such rare situations and ensure the soundness of our analysis, we also report additional metrics. We define two groups of metrics to holistically evaluate different aspects of a trained super-net. The first

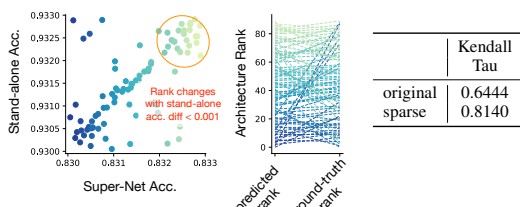

Figure 3: **Kendall-Tau vs sparse Kendall-Tau.** Kendall-Tau is not robust when many architectures have similar performance. Minor performance differences can lead to large perturbations in the ranking. Our sparse Kendall-Tau alleviates this by dismissing minor differences in performance.

group of metrics directly evaluates the quality of the super-net, including sparse Kendall-Tau and the widely-adopted super-net accuracy. For the super-net accuracy, we report the average accuracy of 200 architectures on the validation set of the dataset of interest. We will refer to this metric simply as *accuracy*. It is frequently used (Guo et al., 2019; Chu et al., 2019) to assess the quality of the trained super-net, but we will show later that it is in fact a poor predictor of the final stand-alone performance. The metrics in the second group evaluate the search performance of a trained super-net. The first metric is the probability to surpass random search: Given the ground-truth rank $r$ of the best architecture found after $n$ runs and the maximum rank $r_{max}$, equal to the total number of architectures, the probability that the best architecture found is better than a randomly searched one is given by $p = 1 - (1 - (r/r_{max}))^n$. Finally, where appropriate, we report the stand-alone accuracy of the model that was found by the complete WS-NAS algorithm. Concretely, we randomly sample 200 architectures, select the 3 best models based on the super-net accuracy and query the ground-truth performance. We then take the mean of these architectures as stand-alone accuracy. Note that the same architectures are used to compute the sparse Kendall-Tau.

## 4 ANALYSIS

We provide an analysis on the impact of the factors that are shown in Table 1 across three different search spaces. Note that, in this section, we present the factors that are the most important for performance; our analysis of the remaining factors is provided in Appendix C.

### 4.1 EVALUATION OF A SUPER-NET

The standalone performance of the architecture that is found by a NAS algorithm is clearly the most important metric to judge its merits. However, in practice, one cannot access this metric—we wouldn't need NAS if standalone performance was easy to query (the cost of computing stand-alone performance is discussed in Appendix B.2). Furthermore, stand-alone performance inevitably depends the sampling policy, and does not directly evaluate the quality of the super-net (see Appendix B.3). Consequently, it is important to rely on metrics that are well correlated with the final performance but can be queried efficiently. To this end, we collect all our experiments and plot the pairwise correlation between final performance, sparse Kendall-Tau, and super-net accuracy. As shown in Figure 4, the super-net accuracy has a low

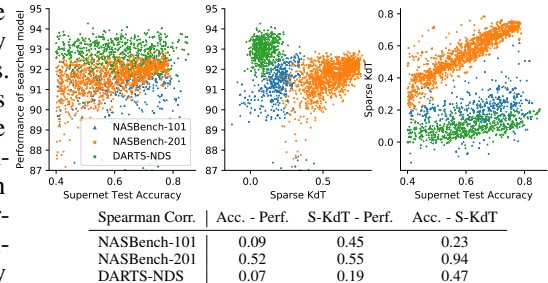

| Spearman Corr. | Acc. - Perf. | S-KdT - Perf. | Acc. - S-KdT |
|---|---|---|---|
| NASBench-101 | 0.09 | 0.45 | 0.23 |
| NASBench-201 | 0.52 | 0.55 | 0.94 |
| DARTS-NDS | 0.07 | 0.19 | 0.47 |

Figure 4: **Super-net evaluation**. We collect all experiments across 3 benchmark spaces. **(Top)** Pairwise plots of super-net accuracy, final performance, and the sparse Kendall-Tau. Each point corresponds to statistics computed over a trained super-net. **(Bottom)** Spearman correlation coefficients between the metrics.

correlation with the final performance on NASBench-101 and DARTS-NDS. Only on NASBench-201 does it reach a correlation of 0.52. The sparse Kendall-Tau yields a consistently higher correlation with the final performance. This is evidence that one should not focus too strongly on improving the super-net accuracy. While this metric remains computationally heavy, it serves as a middle ground that is feasible to evaluate in real-world applications.

In the following experiments, we thus mainly rely on sparse Kendall-Tau, and use final search performance as a reference only. We report the training details in Appendix B.1 and the complete results of all metrics in Appendix C.6.

## 4.2 BATCH NORMALIZATION IN THE SUPER-NET

Batch normalization (BN) is commonly used in standalone networks to allow for faster and more stable training. It is thus also employed in most CNN search spaces. However, BN behaves differently in the context of WS-NAS, and special care has to be taken when using it. In a standalone network (c.f. Figure 5 *(Top)*), a BN layer during training computes the batch statistics $\mu_B$ and $\sigma_B$, normalizes

the activations $f_A(x)$ as $(f_A(x) - \mu_B)/\sigma_B$, and finally updates the population statistics using a moving average. For instance, the mean statistics is updated as $\hat{\mu} \leftarrow \gamma\hat{\mu} + (1-\gamma)\mu_B$. At test time, the stored population statistics are used to normalize the feature map. In the standalone setting, both batch and population statistics are unbiased estimators of the population distribution $\mathcal{N}(\mu, \sigma)$.

By contrast, when training a super-net (Figure 5 *(Bottom)*) the population statistics that are computed based on the running average are not unbiased estimators of the population distribution, because the effective architecture before the BN layer varies in each epoch. More formally, let $f_{A_i}$ denote the $i$-th architecture. During training, the batch statistics are computed as $\mu_B^i = \sum_j f_{A_i}(x_j)/m$, and the output feature follows the distribution $\mathcal{N}(\mu_B^i, \sigma_B^i)$, where the superscript $i$ indicates that the current batch statistics depends on $A_i$ only. The population mean statistics is then updated as $\hat{\mu} \leftarrow \gamma\hat{\mu} + (1-\gamma)\mu_B^i$. However, during training, different architecture from the super-net are sampled. Therefore, the population mean statistics essentially becomes a weighted combination of means from different architectures, i.e., $\hat{\mu} \leftarrow \sum \alpha_i \mu_B^i = \sum \alpha_i f_{A_i}(x)$, where $\alpha_i$ is the sampling frequency of the $i$-th architecture. When evaluating a specific architecture $A_i$ at test time, the estimated population statistics thus depend on the other architectures in the super-net. This leads to a train-test discrepancy. One solution to mitigate this problem is to re-calibrate the batch statistics by recomputing the statistics on the entire training set before the the final evaluation (Yu & Huang, 2019). While the cost of

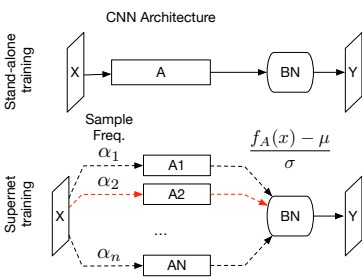

Figure 5: Batch normalization in standalone and super-net training.

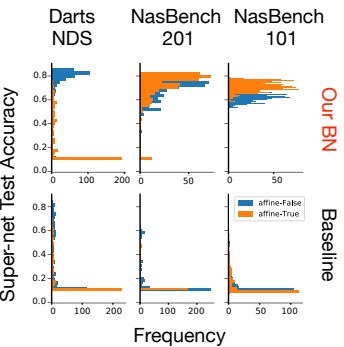

Figure 6: Validation of BN.

doing so is negligible for a standalone network, NAS algorithms typically sample $\sim 10^5$ architectures for evaluation, which makes this approach intractable.

In contrast to Dong & Yang (2020) and Bender et al. (2020) who use the training mode also during testing, we formalize a simple, yet effective, approach to tackle the train-test discrepancy of BN in super-net training: we leave the normalization based on batch statistics during training unchanged, but use batch statistics also during testing. Since super-net evaluation is always conducted over a complete dataset, we are free to perform inference in mini-batches of the same size as the ones used during training. This allows us to compute the batch statistics on the fly in the exact same way as during training.

Figure 6 compares standard BN to our proposed modification. Using the tracked population statistics leads to many architectures with an accuracy around 10%, i.e., performing no better than random guessing. Our proposed modification allows us to significantly increase the fraction of

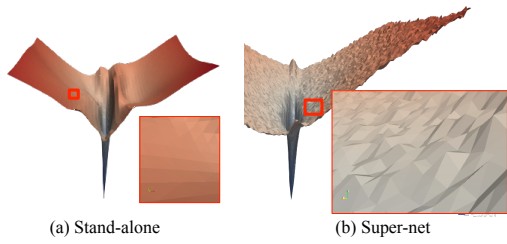

Figure 7: Loss landscapes.

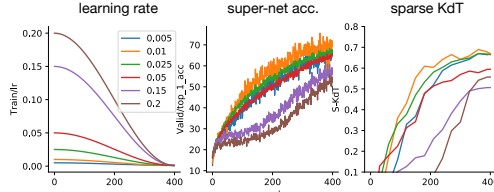

Figure 8: Learning rate on NASBench-201.

high-performing architectures. Our results also show that the choice of fixing vs. learning an affine transformation in batch normalization should match the standalone protocol $P_{proxy}$.

### 4.3 SUPER-NET LOSS LANDSCAPES

The training loss of the super-net encompasses the task losses of all possible architectures. We suspect that the training difficulty increases with the number of architectures represented by the super-net. To better study this, we visualize the loss landscape (Li et al., 2018) of the standalone network and a super-net with $n = 300$ architectures. Concretely, the landscape is computed over the super-net training loss under the single-path one-shot sampling method,

$$\mathcal{L}_s(x, \theta_s) = \sum_i \mathcal{L}_s(x, \theta_i), \quad \text{where } \forall i, \cup_i \theta_i = \theta_s. \tag{1}$$

Figure 7 shows that the loss landscape of the super-net is less smooth than that of a standalone architecture, which confirms our intuition. A smoother landscape indicates that optimization will converge more easily to a good local optimum. With a smooth landscape, one can thus use a relatively large learning. By contrast, a less smooth landscape requires using a smaller one.

Our experiments further confirm this observation. In the standalone protocol $P_{proxy}$, the learning rate is set to 0.2 for NASBench-101, and to 0.1 for NASBench-201 and DARTS-NDS, respectively. All protocols use a cosine learning rate decay. Figure 8 shows that super-net training requires lower learning rates than standalone training. The same trend is shown for other search spaces in Appendix C.1. We set the learning rate to 0.025 to be consistent across the three search spaces.

### 4.4 LOWER FIDELITY ESTIMATES LOWER THE RANKING CORRELATION

Reducing memory foot-print and training time by proposing smaller super-nets has been an active research direction, and the resulting super-nets are referred to as *lower fidelity estimates* (Elsken et al., 2019). The impact of this approach on the super-net quality, however, has never been studied systematically over multiple search spaces . We compare four popular strategies in Table 2. We deliberately prolong the training epochs inversely proportionally to the computational budget that would be saved by the low-fidelity estimates, e.g. if the channel number is reduced by half, we train the model for two times more epoch. Note that this provides an upper bound to the performance of low-fidelity estimates.

A commonly-used approach to reduce memory requirements is to decrease the batch size (Yang et al., 2020). Surprisingly, lowering the batch size from 256 to 64 has limited impact on the accuracy, but decreases sparse Kendall-Tau and the final searched model's performance, the most important metric in practice.

Table 2: Low fidelity estimates under same computational budget, reporting final search model accuracy (FSA) and sparse Kendall-Tau (S-KdT) on NASBench-201.

| Metrics | Settings | | |
|---|---|---|---|
| Repeated cells | 3 | 2 | 1 |
| S-KdT | $0.751 \pm 0.09$ | $0.692 \pm 0.18$ | $0.502 \pm 0.21$ |
| FSA | $91.91 \pm 0.09$ | $91.95 \pm 0.10$ | $90.30 \pm 0.71$ |
| Init Channel | 16 | 8 | 4 |
| S-KdT | $0.740 \pm 0.07$ | $0.677 \pm 0.10$ | $0.691 \pm 0.15$ |
| FSA | $92.92 \pm 0.48$ | $92.32 \pm 0.37$ | $92.79 \pm 0.85$ |
| Batch-size | 256 | 128 | 64 |
| S-KdT | $0.740 \pm 0.07$ | $0.728 \pm 0.16$ | $0.703 \pm 0.16$ |
| FSA | $92.92 \pm 0.48$ | $92.37 \pm 0.61$ | $92.35 \pm 0.34$ |
| Train portion | 0.75 | 0.5 | 0.25 |
| S-KdT | $0.751 \pm 0.11$ | $0.742 \pm 0.12$ | $0.693 \pm 0.13$ |
| FSA | $92.13 \pm 0.51$ | $92.74 \pm 0.43$ | $91.47 \pm 0.81$ |

Another approach is to decrease the number of channels in the first layer (Liu et al., 2019b). This reduces the total number of parameters proportionally, since the number of channels in consecutive

| Type | Accuracy | S-KdT | P > R | Final searched model |
|---|---|---|---|---|
| Fixed | $71.52 \pm 6.94$ | 0.22 | 0.546 | $91.79 \pm 1.72$ |
| Shuffle | $31.79 \pm 10.90$ | 0.17 | 0.391 | $90.58 \pm 1.58$ |
| Interpolate | $57.53 \pm 10.05$ | 0.37 | 0.865 | $93.35 \pm 3.27$ |
| Baseline† | $76.91 \pm 10.05$ | 0.22 | 0.865 | $89.43 \pm 4.30$ |
| Baseline-v2 | $75.18 \pm 9.28$ | 0.33 | 0.891 | $91.27 \pm 1.18$ |
| Ours | $\mathbf{76.95} \pm 8.29$ | **0.46** | **0.949** | $\mathbf{93.65} \pm 0.73$ |

† See Appendix C.3 for more details.

Table 3: Dynamic channels on NASBench-101.

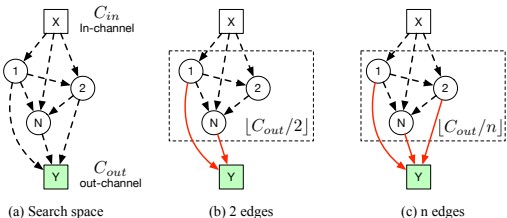

(a) Search space     (b) 2 edges     (c) n edges

Figure 9: NASBench-101 dynamic channel.

layers depends on the first one. Table 2 shows that this decreases the sparse Kendall-Tau from 0.7 to 0.5. By contrast, reducing the number of repeated cells (Pham et al., 2018; Chu et al., 2019) by one has little impact. Hence, to train a good super-net, one should avoid changes between $f_{ws}$ and $f_{proxy}$, but one can reduce the batch size by a factor $> 0.5$ and use only one repeated cell.

The last lower-fidelity factor is the portion of training data that is used (Liu et al., 2019b; Xu et al., 2020). Surprisingly, reducing the training portion only marginally decreases the sparse Kendall-Tau for all three search spaces. On NASBench-201, keeping only 25% of the CIFAR-10 dataset results in a 0.1 drop in sparse Kendall-Tau. This explains why DARTS-based methods typically use only 50% of the data to train the super-net but can still produce reasonable results.

### 4.5 DYNAMIC CHANNELING HURTS SUPER-NET QUALITY

Dynamic channeling is an implicit factor in many search spaces (Ying et al., 2019; Cai et al., 2019; Guo et al., 2019; Dong & Yang, 2019b). It refers to the fact that the number of channels of the intermediate layers depends on the number of incoming edges to the output node. This is depicted by Figure 9 *(a)*: for a search cell with $n$ intermediate nodes, where $X$ and $Y$ are the input and output node with $C_{in}$ and $C_{out}$ channels, respectively. When there are $n = 2$ edges (c.f. Figure 9 *(b)*), the associated channel numbers decrease so that their sum equals $C_{out}$. That is, the intermediate nodes have $\lfloor C_{out}/2 \rfloor$ channels. In the general case, shown in Figure 9 *(c)*, the number of channels in intermediate nodes is thus $\lfloor C_{out}/n \rfloor$ for $n$ incoming edges. A weight sharing approach has to cope with this architecture-dependent fluctuation of the number of channels during training.

Let $C$ denote the number of channels of a given architecture, and $C_{max}$ the maximum number of channels for a node across the entire search space. All existing approaches allocate $C_{max}$ channels and, during training, extract a subset of these channels. The existing methods then differ in how they extract the channels: Guo et al. (2019) use a fixed chunk of channels, e.g., $[0 : C]$; Zhang et al. (2018) randomly shuffle the channels before extracting a fixed chunk; and Dong & Yang (2019a) linearly interpolate the $C_{max}$ channels into $C$ channels using a moving average across neighboring channels.

Instead of sharing the channels between architectures, we propose to disable dynamic channelling completely. As the channel number only depends on the incoming edges, we separate the search space into a discrete number of sub-spaces, each with a fixed number of incoming edges. As shown in Table 3, disabling dynamic channeling improves the sparse Kendall-Tau and the final search performance by a large margin and yields a new state of the art on NASBench101.

We compose another baseline, where we enable dynamic channeling during super-net training. During validation, we compute the average sparse Kendall-Tau of each sub-space, where we sample 200 architectures that shares the same number of channels. We call this baseline-v2. In Table 3, we can see this surpasses the original baseline by a significant margin. It further evidence the importance of disabling dynamic channels. Nonetheless, the best is to disable dynamic channeling during both the training and the validation phase.

## 5 HOW SHOULD YOU TRAIN YOUR SUPER-NET?

Figure 10 summarizes the influence of all tested factors on the final performance. It stands out that properly tuned hyper-parameters lead to the biggest improvements by far. Surprisingly, most other factors and techniques either have a hardly measurable effect or in some cases even lead to worse performance. Based on these findings, here is how you should train your super-net:

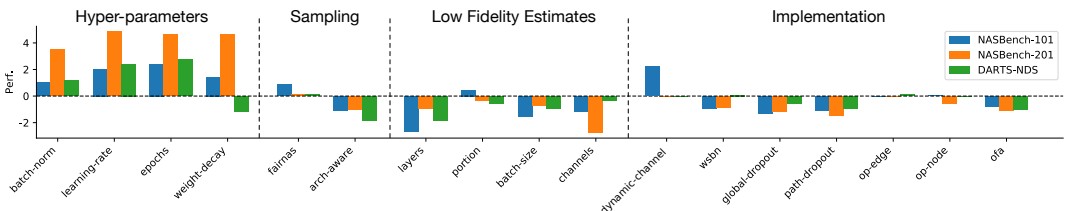

Figure 10: **Influence of factors on the final model.** We plot the difference in percent between the searched model's performance with and without applying the corresponding factor. For the hyper-parameters of $P_{ws}$, the baseline is Random NAS, as reported in Table 4. For the other factors, the baseline of each search space uses the best setting of the hyper-parameters. Each experiment was run at least 3 times.

1. Do not use super-net accuracy to judge the quality of your super-net. The sparse Kendall-Tau has much higher correlation with the final search performance.
2. When batch normalization is used, do not use the moving average statistics during evaluation. Instead, compute the statistics on the fly over a batch of the same size as used during training.
3. The loss landscape of super-nets is less smooth than that of standalone networks. Start from a smaller learning rate than standalone training.
4. Do not use other low-fidelity estimates than moderately reducing the training set size to decrease the search time.
5. Do not use dynamic channeling in search spaces that have a varying number of channels in the intermediate nodes. Break the search space into multiple sub-spaces such that dynamic channeling is not required.

**Comparison to the state of the art.** Table 4 shows that carefully controlling the relevant factors and adopting the techniques proposed in Section 4 allow us to considerably improve the performance of Random-NAS. Thanks to our evaluation, we were able to show that simple Random-NAS together with an appropriate training protocol $P_{ws}$ and mapping function $f_{ws}$ yields results that are competitive to and sometimes even surpass state-of-the-art algorithms. Our results provide a strong baseline upon which future work can build.

Table 4: **Final results.** Results on NASBench-101 and 201 are from Yu et al. (2020b), and Dong & Yang (2020). We report the mean over 3 runs. Note that NASBench-101 ($n = 7$) in (Yu et al., 2020b) is identical to our setting. Our new strategy significantly surpasses the random search baseline.

| Method | NASBench 101 (n=7) | NASBench 201 | DARTS NDS | DARTS NDS* |
|---|---|---|---|---|
| ENAS | $91.83 \pm 0.42$ | $54.30 \pm 0.00$ | $94.45 \pm 0.09$ | 97.11 |
| DARTS-V2 | $92.21 \pm 0.61$ | $54.30 \pm 0.00$ | $94.79 \pm 0.11$ | 97.37 |
| NAO | $92.59 \pm 0.59$ | - | - | 97.10 |
| GDAS | - | $93.51 \pm 0.13$ | - | 96.23 |
| Random NAS | $89.89 \pm 3.89$ | $87.66 \pm 1.69$ | $91.33 \pm 0.12$ | $96.74^{\dagger}$ |
| Random NAS (Ours) | $93.12 \pm 0.06$ | $92.71 \pm 0.15$ | $94.26 \pm 0.05$ | 97.08 |

$^{\dagger}$Results from Li & Talwalkar (2019)
*Trained according to Liu et al. (2019b) for 600 epochs.
DARTS-V2 (Liu et al., 2019b), ENAS (Pham et al., 2018), NAO (Luo et al., 2018).
Random-NAS (Li & Talwalkar, 2019), GDAS (Dong & Yang, 2019b)
On NASBench-201, both random NAS and our approach samples 100 final architectures to follow Dong & Yang (2020)

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
