# OpenReview forum: "How to Train Your Super-Net: An Analysis of Training Heuristics in Weight-Sharing NAS"
_ICLR.cc/2021/Conference — Reject_

### Official Review · AnonReviewer2 · 2020-10-17
**Paper Review**

**Rating:** 5
**Confidence:** 3

**Review:**

## Summary
Neural Architecture Search (NAS) aims to find a model with the best possible accuracy (or best possible accuracy/size tradeoff) from within a human-defined search space. One popular strategy for speeding up NAS is to train a one-shot model -- a single set of shared weights -- that can then be used to evaluate any candidate architecture within the search space without retraining or fine-tuning. The submission investigates how various decisions made when training a one-shot model can affect its ability to rank different candidate architectures within a search space.

The submission presents an empirical study, and I believe its main contribution is to provide information about (i) how sensitive one-shot model quality is to hyper-parameter tuning, and (ii) how to tune the hyper-parameters of a one-shot model in order to produce high-quality search results. These results are likely to be of practical interest to people trying to train high-quality one-shot models for neural architecture search, as well as for researchers trying to benchmark and compare different NAS algorithms.

The submission presents several sets of experiments. The most significant set of experiments (in my opinion) is Figure 11 ("influence of factors on the final model"), which shows the effect on the quality of the final searched model when we tune different hyper-parameters of the one-shot model, such as the learning rate, number of training epochs, weight decay, sampling scheme, and batch norm implementation, or apply various heuristics for decreasing the training time and memory usage of the one-shot model. Several additional sets of experiments are presented throughout Sections 4 and 5 of the paper.

In general, the paper is clear and well-organized, and most of the experiments are well-described; two relatively minor exception are (i) Figure 1, which I had trouble making sense of initially, and (ii) some of the experiments on the right-hand side of Figure 11 (e.g., "ofa kernel," "op-edge"), which include citations but where the authors' precise setup doesn't seem to be clearly defined in the main paper.

**Pros:**
* Conducts a broad set of experiments evaluating the effect of different training decisions on the quality of a one-shot model.
* The experiments and concrete recommendations made by the paper are likely to be of practical value for practitioners/researchers who want to train high-quality one-shot models.
* Experiments reported on a variety of different search spaces/benchmark tasks.

**Cons:**
* Experiments are limited to CIFAR-10.
* Some unresolved questions about a few of the experiments. (Discussed below.)

**Notes on Paper Rating**: I have some concerns about how a few of the experiments are presented  (detailed below) which would need to be addressed before I felt comfortable advocating for the paper's acceptance. If they are adequately addressed, however, I think the paper could be a valuable resource for NAS practitioners.

## Notes on Empirical Evaluation
The submission conducts experiments on (certain subsets of) the NASBench-101 and NASBench-201 benchmark tasks, as well as a third task (attributed to Radosavovic et al., 2019) which the submission refers to as DARTS-NDS. One caveat is that all the conclusions in the paper appear to be drawn based on experiments on the CIFAR-10 image classification dataset. While it's unclear how well the results would generalize to other datasets, I think this is an acceptable limitation, given (i) the broad set of experiments conducted on this dataset, and (ii) most existing NAS Benchmark tasks I'm aware of are built around CIFAR-10.

At various points in the paper, the quality of the one-shot model is measured according to one of three different metrics: (i) "sparse Kendall Tau", which measures the correlation between accuracies measured according to the one-shot model and ground-truth accuracies obtain by training network architectures from scratch, (ii) the ground-truth accuracy of the network architecture which is ranked highest according to the one-shot model, and (iii) the probability that a network architecture found using a one-shot model will be better than one found using a random search baseline without weight sharing. What this means in practice is that different sets of experiments use different metrics. This leads to a slightly inconsistent presentation. For example, the experiments in Section 4.4 ("lower fidelity estimates") focus on Sparse Kendall Tau, while the experiments in Section 5 / Figure 11 ("Influence of factors on the final model") focus on ground-truth accuracies. However, experiments within the same section/plot are typically comparable to each other.

The weakest part of the paper is Table 3 ("Final Results") which compares the results of the authors' well-tuned one-shot model against previously published results. The raw numbers are compelling, but I had trouble verifying the comparison between the "Random NAS" and "Random NAS (Ours)" rows of the table. In particular:
1. The 87.66 accuracy number for "Random NAS" in NASBench-201 was obtained by sampling 100 random architectures from the search space and retaining only the one with the highest one-shot model accuracy. It was unclear to me whether "Random NAS (Ours)" used the same evaluation protocol.
2. It was unclear to me how the 91.33 accuracy number for DARTS-NDS Random NAS was obtained, since I was unable to find that number when I skimmed Radosavovic's paper.
3. If I understand the table caption correctly, the "NASBench-101" table header is misleading and should be changed to "NASBench-101 (n=7)".
**Because the improvement of "Random NAS (Ours)" over "Random NAS" is highlighted in Section 5, these points would need to be addressed before I felt comfortable recommending the paper for acceptance.**

## Additional Notes
In Section 4.4: It would be helpful to explicitly state how the learning rate is adjusted when you decrease the batch size, since this can significantly affect the final results.

In Section 4.5: It would be helpful to update the submission to clarify: if you disable dynamic channels and train a separate model for each possible number of incoming edges, do you include comparisons between candidate architectures with different numbers of incoming edges when you compute the sparse Kendall-Tau? (If not, it seems like the sparse Sparse Kendall Tau numbers for different methods might not be directly comparable to each other.)

---

> ### Author Response · Authors · 2020-11-24
> **Answer to your questions**
>
> Thank you for the valuable suggestions and for acknowledging our contribution. Here are our replies to your specific questions.
>
> Q1: Metrics to evaluate super-net
> We revised Section 4.4 to include both sparse Kendall Tau and final accuracy. Fundamentally, sparse Kendall Tau and super-net accuracy are metrics one can compute during super-net training to evaluate the quality. On the other hand, the final searched model accuracy is associated with a particular NAS sampler and can be computed after the search is finished. For completeness, we also report the full metrics in supplementary materials for all experiments.
>
> Q2: Table 3 final results.
> We updated the RandomNAS(Ours) results on NASBench-201 to match the RandomNAS baseline, sampling 100 architectures instead of 200. We observed that doing so only marginally decreases the performance of our model and does not affect our conclusions.
> 91.33 is an accuracy that we obtained in our experiments; the previous papers do not report RandomNAS results. We added the standard deviation and clarified this in the main paper.
> We revised the table to match NASBench-101 (n=7). Nonetheless, this refers to the entire NASBench-101 space with 423K architectures in total.
>
> Q3: tuning learning rate when decreasing the batch size
> Due to limited time and resources, we had to postpone such a comparison to future work.
>
> Q4: Section 4.5, dynamic channeling
> Thanks for the suggestion. We constructed another baseline by training with dynamic channels but computing the sparse Kendall Tau for each sub-space, where the architectures do not have dynamic channels. While this indeed improves the S-KdT of the baseline from 0.23 to 0.33, our method still surpasses this new baseline, with an S-KdT of 0.46. Note that both the new baseline and our results indicate that, when using dynamic channeling, one should disable it in both the training and testing phases.

---

### Official Review · AnonReviewer3 · 2020-10-29
**[Initial Rating -- Borderline] Lack evidence on large-scale dataset**

**Rating:** 5
**Confidence:** 5

**Review:**

This paper aims to analyze 14 commonly used training tricks for weight-sharing based NAS algorithms. The empirical analysis is evaluated on three existing NAS datasets.

Pros:
- A comprehensive analysis of commonly used NAS training tricks.
- A new metric -- sparse Kendall-Tau -- for the correlation between architectures.

Cons:
- No new ideas are proposed, as the authors are evaluating existing tricks.
- Most of the conclusions are somehow already revealed in previous NAS works.
- As an emprical analysis paper, the authors do have some interesting finds. However, could these finds be used to improve existing NAS algorithms (which mostly already integrate multiple different training tricks)? A more concrete example is could the authors use the proposed findings to outperform the recent TuNAS?
- The authors claimed to evaluate and analyse 14 factors, and use one page to explain the behavior of BN in Sec.4.2. However, "the proposed modification" is already mentioned and analyzed in previous papers, such as https://arxiv.org/abs/2001.00326 and https://arxiv.org/abs/2008.06120.
- The explanation in Sec.4.3 is unclear to me. The learning rate schedule does not consider the commonly used warmup strategy. From Figure 8, it seems 0.01 results in the best correlation, but the authors choose 0.025 without further explanation. The authors mentioned "requires a smaller lr", whereas 0.005 is worse than 0.01. Overall speaking, in Sec.4.3, the loss landscapes looks interesting to me, but other analysis is not convincing to me.
- All the experiments in this paper focus on the DARTS-like search space. There are also some other popular search spaces, such as MobileNet-like search space. It remains a question on how does the conclusion generalize to other search spaces.


Minor issues:
- In Figure 1, to make it self-contained, I would suggest to explain P_{ws}, f_{xxx} in the caption.
- In Figure 1, "NAS-Bench-102" -> "NAS-Bench-201"
- In Figure 1, why use "Proxy-task Training protocol" for the evaluation phase? In my opinion, the search is to proxy the evaluation task?
- Regarding the presentation, f_{xxx} and P_{xxx} in Sec 2 take me a lot of time to understand. If the authors could add some examples when introducing these concepts, it could help readers understand.
- In Sec.4.4, "The impact of this approach on the super-net quality, however, has never been studied." is somewhat arbitrary. The results of #cells and #channels for DARTS is reported in their rebuttal, and I believe many researchers try these hyperparameters in their experiments but just not mentioned in the paper.
- In Sec.4.5, the reference before "linearly interpolate" should be "Network Pruning via Transformable Architecture Search" instead of the cited one.
- "SNAS" is accepted by ICLR 2019, "FBNet" is accepted by CVPR,  "Auto-DeepLab", etc; I would like to suggest authors cite their official version instead of arxiv version.
- I'm confused about "disabling dynamic channeling". Would the authors mind to explain more or share the codes on how to do it? In addition, it seems this strategy can only be used for NAS-Bench-101 search space, instead of others.

Some questions:
- I'm confused about Figure 4 - 2/3. How could the x-axis be "sparse KdT"? If x-axis is "sparse KdT", it means each data architecture has a "sparse KdT" value? But if I understand correctly, "sparse KdT" is evaluated on a set of pairs, instead of a single data.
- In Sec.3.2, why does less smoothed loss landscape indicate a smaller learning rate?
- Would the authors mind adding the results of #channels=32 / #batch size=32/512 / #cells=1,5 in Figure 9?


**Post Rebuttal Comments**:
I appreciate the authors' efforts on the rebuttal, most of my questions are answered and addressed. However, I still have a major concerns. As agreed by the authors' response, this paper lies in a detailed analysis of existing tricks. I thus hope the paper could bring some new insights, but the mentioned "other important conclusions" in response is somehow commonly known in the NAS community. This makes the paper like some empirical supplementary material for existing papers instead of a new one. Therefore, I keep my original rating.

---

> ### Author Response · Authors · 2020-11-24
> **Answers to "Cons" -- We aim to have a better understanding of each factor of the super-net**
>
> Thank you very much for your detailed suggestions. We revised our paper to incorporate them. Here are our replies to your major concerns.
>
> Q1. No new ideas are proposed, author evaluates existing tricks.
> Our contribution lies in the detailed analysis in a fair, controlled environment. We believe such analysis is critical in the current NAS literature as this field advances so fast and all the important information is scattered in different settings. As you also mentioned, some of the prior works, e.g., DARTS, report a partial hyper-parameter tuning for their own algorithm in their supplementary material. Others, e.g., NASBench-201 and TuNAS, briefly mention batch-normalization without any justification. This is exactly what our work addresses: We collect factors that are scattered throughout the literature and clearly analyze them in a systematic fashion to provide the community a rigorous understanding of which factors are useful and which ones are not.
>
> Furthermore, our comprehensive analysis allowed us to draw other important conclusions: 1) Super-net accuracy should not be trusted and one should rather rely on the sparse Kendall Tau; 2) low fidelity estimates should be used cautiously.
>
> Ultimately, we therefore believe that our work brings a significant contribution to the NAS community and will be highly valuable to the researchers in this field.
>
>
> Q2. Most of the conclusions somehow revealed in previous NAS works.
> Previous works use low-fidelity estimates to reduce the computational cost, but in Section 4.4 we show that, in fact, the super-net quality will be negatively impacted.
> The study of the loss landscape was never performed before, and thus no previous work identified the need to use smaller learning rates.
> We explicitly show the benefits of using sparse Kendall tau over super-net accuracy to evaluate the super-net quality.
> The influence of dynamic channels was never shown before.
>
> Q3 Improving the existing NAS algorithm
> TuNAS is based on reinforcement learning. We show the improvement in Random-NAS, which is one of the most widely used NAS approaches. Nevertheless, this is not the main focus of this paper, we aim to have a better understanding of each factor of the super-net, not to tune the hyper-parameters of a specific algorithm.
>
> Q4 BN behavior.
> Thanks for pointing out these papers. In NASBench201 (https://arxiv.org/abs/2001.00326), Dong and Yang mentioned that they dropped the batch-statistics without concrete analysis; this is also the case in TuNAS (https://arxiv.org/abs/2008.06120). In fact, this was only added to the NASBench-201 paper in a revised version in Jan 2020, and we based our work on the Nov 2019 version; furthermore, TuNAS only mentions this in appendix E.2 in one sentence.
>
> In contrast to these works, we clearly analyze why this adaption should be made in super-net batch normalization. Nevertheless, we will tune down our claim, and cite these works to give proper credit.
>
> Q5 Learning rate
> We advocate for a smaller learning rate compared to the stand-alone learning rate. This does not mean that the learning rate should be as small as possible. We clarified this in the paper.
>
> Instead of picking the optimal learning rate for each search space, we set it to 0.025 to be consistent across the three search spaces while obtaining similar Sparse Kendall Tau performance as the configurations you mentioned.
>
> Q6 DARTS-like search space
> Unfortunately, as mentioned by R2, benchmark datasets with a sufficient amount of architectures with known stand-alone performance are only available for DARTS-like (or in general NASNet-like) search spaces. Nonetheless, the DARTS search space is one of the most important search spaces in NAS and frequently used in the literature. More importantly, it is a general search space where one searches for both the topology of the architecture and the individual operations.  This is much harder than searching in linear MobileNet-like spaces, where one only searches for convolutional kernel sizes or channel ratios. Working in more complex search spaces is crucial to make progress towards a truly automatic architecture search.

---

> ### Author Response · Authors · 2020-11-24
> **Replies to ‘some questions’ and minor issues**
>
>
> Q1. Figure 4, (2) and (3) sparse KdT in the x-axis
> As stated in the caption, each data point represents the statistics obtained over a trained super-net. It does not represent architecture. These plots were generated by gathering all of our experiments in each search space.
>
> Q2 Loss landscape.
> A smoother landscape indicates that optimization will converge more easily to a good local optimum. With a smooth landscape, one can thus use a relatively large learning rate. By contrast, a less smooth landscape requires using a smaller one.
>
> Q3: many researchers report their hyper-parameters tuning
> We revise the original statement. This is actually why our study is important; previous works only report partial hyper-parameters on one specific search space and for one specific algorithm. By contrast, we study all factors in a systematic fashion over multiple spaces and in a controlled environment.
>
> Q4 Adding more configurations for section 4.4.
>
> For #channel = 32, as stand-alone network in NASBench-201 has #channels =16, Increasing this channel number to 32 violates the concept of low fidelity estimates.
>
> #cells = 5 is the baseline, while #cells = 1 is now reported in Table 2 in the revised paper. #batch-size = 32 will have a S-KdT of 0.701 and final search performance of 92.10.
>
> Q5: disable dynamic channeling
> As stated in Section 4.5, “As the channel number only depends on the incoming edges, we separate the search space into a discrete number of sub-spaces, each with a fixed number of incoming edges.” So no dynamic channeling is used in each individual sub search space.

---

### Official Review · AnonReviewer4 · 2020-10-30
**Analysis of Supernet Training Process for NAS**

**Rating:** 5
**Confidence:** 4

**Review:**

Summary:
The paper analyzes the effect of different components of the supernet training process w.r.t performance of the final NAS models. Finally it makes some recommendations for good practices for training supernets that eventually lead to better NAS designed models.

Strengths:
+ The performance of many NAS approaches is critically dependent on the effectiveness of Supernets. As such, a thorough study of supernet training process is timely and useful.
+ The final recommendations are useful, albeit unsurprising. Standalone or supernet, training hyper-parameters usually play a big role.
+ The discussion on batch norm is helpful. This is indeed a problem with some kinds of supernet training methods.

Weaknesses:
- I did not find information on the supernet training process that is used. There are many supernet training method proposed in the literature but it is not clear what is similar or different in the paper training process. I looked for this but could not find it, it is possible I missed it, but this seems to be important information that should not be hard to find.
- As existing supernet models (e.g., OFA, SPOS, BigNAS) show, sampled subnets can be directly used, with the same level of accuracy compared to training from scratch. This seems to run counter to the main premise of the current paper, which is that the supernet only provides an architecture, which then needs to be trained properly. As such it is not clear how much of the analysis in this paper is applicable to current supernets, and consequently how useful this analysis is.
- Following from the above, the OnceForAll supernet is one of the most effective one, which does incorporate some of the suggestions from this paper. There does not seem to be much discussion of the OFA supernet apart from the OFA conv.
- Parts of the paper are very confusing and not easy to understand. As an example, in section 2, $f_{ws}$ is introduced as a mapping but not defined until much later.

Overall, the paper analyzes the effect of different components of the supernet training process. While parts of the paper are interesting, it does not discuss the training process of some of the latest and most effective supernet models. As such it is not clear how useful the findings  of this paper are.

---

> ### Author Response · Authors · 2020-11-24
> **Revised our paper to address your concerns**
>
> Thanks for your suggestions. We have revised our paper accordingly. Here are replies to your specific questions.
>
> Q1: Supernet training process.
> We mention the super-net training under “Section 3.1 Weight-sharing protocol”. We revised this section to make the training process clearer.
>
> Q2: SOTA OFA / BigNAS shows high super-net accuracy on MobileNet-like space.
> MobileNet-like search spaces are only one of many existing search spaces. This search space is effective to search for a compact model on mobile devices. However, it has a simplistic and well-behaved design, as it only searches for the sizes of convolutional kernels and the number of feature maps. In the Mobile-Net like linear search spaces, the slimmable networks (Yu et al. 2018) showed that sharing the weights improves the super-net accuracy. The follow-up works  BigNAS (Yu et al. 2020) and Once for all (OFA) (Cai et al. 2020) then evidenced that super-net performance can surpass stand-alone training in such linear spaces. However, our experiments clearly show that this behavior cannot be observed in more complex, cell-based search spaces, where the topology and operations are searched jointly.
>
>
> Q3: Lack of OFA discussion.
> Thanks for the suggestion, we revised the paper to include a discussion on OFA.
> In short, the OFA paper targets a MobileNet-like search space, which consists of linear connected layers. For each layer, one can search for the convolutional kernel size {3, 5, 7}, the channel width ratio {3, 6}, and an elastic depth by adding skip-connections. Apart from the OFA kernel, which can generalize to the existing three benchmark search spaces, other approaches used in OFA, i.e. elastic depth or width, do not directly generalize to cell search spaces. This explains why we conducted experiments only with OFA kernels.
>
> Q4: $f_{ws}$.
> We introduce f_proxy as a mapping from architecture encoding to a stand-alone architecture. Following this definition,  f_ws maps an encoding to a shared-network, or, to be more precise, a portion of the shared network. We revised the paper accordingly.

---

### Official Review · AnonReviewer1 · 2020-11-02
**Interesting empirical study of supernet training. Some more information needed and the writing could be improved.**

**Rating:** 5
**Confidence:** 4

**Review:**

This work analyzes commonly used heuristics for training the supernet in weight sharing NAS. The authors first proposes a new metric, sparse Kendall-Tau, to measure the quality of the supernet. Then extensive experiments are conducted on three NAS benchmarks to empirically evaluate the heuristics, and pick the best settings. To highlight the significance of the training quality of supernet, the author showed that random search, when combined with the best settings, can performs competitive to SOTA results.

Strength:

1. The supernet training quality is an important factor in weight sharing NAS, which, as argued in this paper, has not received enough attention. This work raises the concern about this issue and empirically shows that it can indeed affect the search result significantly.

2. The systematic benchmarking of different heuristics is valuable and serves as a useful guide to apply weight sharing NAS effectively.

Weakness:

1. The experiments are all conducted on relatively small search spaces. However, the search spaces in more realistic settings are usually much larger and might have different properties. For example, a recent work (Bender et al, 2020) showed that, contrary to the results on the small search space, random search significantly underperforms weight sharing results on large search space. So it would be interesting to see if the conclusions in this work still holds on large search spaces.

2. The proposed sparse Kendall-Tau metric is shown to be more robust. But it also seems to be more coarse grained. For example, if the differences of top models are small, they would be put into one group by sparse Kd-T thus neglecting their relative ranking. However, those relatively rankings is also important in getting the best search outcome / selecting the best model.

3. It is probably expected that the low-fidelity estimates would make search results worse since they are trading off search quality with efficiency. It would help to calibrate the comparison based on the search budget. For example, the comparisons in Figure 11 should be done while controlling the search budget for each setting and it would be even better to show multiple comparisons under different search budgets.

4. More information about the loss landscape visualization would be appreciated, especially how the losses are computed for the supernet. Since the supernet can be seen as a larger standalone network, it is unintuitive to me why its loss landscape would be so different.

5. The presentation could be improved to make the main message more clear. Some specific points below:

Figure 1 is quite confusing. From the description, it seems the main message is that, from (a) to (c), more and more parts in the whole process are controlled, so it would help to align the elements from different sub-figures to make it easier to compare and see the highlighted differences. Right now, it is a bit cluttered and hard to see the main point. It would also help to simplify the elements. For example, the sub-figures seem to be also describing the differences in the search algorithm's relationship with the other components, but it is not very clear and there isn't any discussion about it.

The use of "the proxy task" seems confusing. From my understanding, this term is usually used to refer to a surrogate task, when the target task is too expensive to search on. An example from (Cai et al., 2018): "they need to utilize~\emph{proxy} tasks, such as training on a smaller dataset, or learning with only a few blocks, or training just for a few epochs. These architectures optimized on proxy tasks are not guaranteed to be optimal on the target task.". However, this paper seems to use "proxy task" to refer to the target task that the NAS is optimizing for. It would help to clarify the difference in the paper.

The term "super-net accuracy" and how it is computed should be defined early on since it is used quite frequently and could be ambiguous to the reader.

Typo:

Appendix B. "This is not a reliable metric, as shown in Fig. 9 in the main paper." ==> I guess it should be "Fig. 8" instead of "Fig. 9" since Fig. 9 is discussing "Low fidelity estimates on NASBench-201".


Bender, Gabriel, et al. "Can Weight Sharing Outperform Random Architecture Search? An Investigation With TuNAS." Proceedings of the IEEE/CVF Conference on Computer Vision and Pattern Recognition. 2020.

Cai, Han, Ligeng Zhu, and Song Han. "Proxylessnas: Direct neural architecture search on target task and hardware." arXiv preprint arXiv:1812.00332 (2018).

---

> ### Author Response · Authors · 2020-11-24
> **Improved our paper according to your suggestions**
>
> Thanks for your suggestions. We have revised our paper accordingly. Here are replies to your specific questions.
>
> Q1: Small search spaces and generalization
> One of our search spaces, DARTS-NDS, encompasses over 3e13 architectures. It may be a misunderstanding as Radosavovic et al. samples only 5,000 architectures to obtain their stand-alone performances. We revised the paper to clarify this.
> TuNAS is parallel to our research. It presents a reinforcement learning sampler with multi-objective search rewards on MobileNet-like search spaces. Extending our analysis to this search space would require a significant amount of pre-trained models. These unfortunately do not exist and would require a very large amount of computing to create. We believe that this goes beyond the scope of our work.
> As shown in Guo et al. 2020 (SPOS), MobileNet-like search spaces are in general simpler because the macro topology is fixed and one only searches for kernel sizes and the number of feature maps. Simple search spaces, while effective for specific applications, limit the general applicability of NAS. By contrast, a NASNet-like cell-based search space supports searching both for cell topology and for a diverse set of operations, which has powered important advances, via methods such as AssembleNet (Ryoo et al. 2019) or AutoDeepLab (Liu et al. 2019).
>
>
> Q2: Sparse Kendall Tau affects final sampling
> Note that sparse Kendall Tau only merges the architectures with a similar loss when computing the ranking correlation. It will not affect the final architecture selection. We still sample the best architecture given its absolute original super-net performance without grouping. Therefore, this problem does not exist.
>
> Q3: Controlled experiments for low-fidelity factors
> Thank you for the suggestion. We prolong the training for those configurations proportionally to ensure the computational budget remains the same for all experiments and revised the paper to present these results. Our results show that training super-nets with low-fidelity estimates for longer does not change the fact that their representation power is reduced. This further supports our initial conclusions.
>
> Q4: Definition of super-net loss landscape.
> To train a network in a stand-alone fashion, the loss is usually defined as $\mathcal{L}(x, \theta)$ where $\theta$ represents the architecture parameters. For a super-net, under the single-path one-shot training regime, this definition is $\mathcal{L}(x, \theta_s) = \sum_i \mathcal{L} (x, \theta_i)$. Thus the super-net loss is a summation over architectures $\theta_i$, where $\forall i, \cup_i \theta_i = \theta_s $, which encompass the super-net parameters. The rest of the loss landscape computation remains the same as for any stand-alone network. We believe that the fact that this illustration runs against intuition is evidence that our findings and evaluations are important.

---

### Author Response · Authors · 2020-11-24
**Paper Revision**

We would like to thank all reviewers for their constructive suggestions, which we incorporated to improve our paper.

Here is an outline of the changes:
- Added a discussion of MobileNet-like search space to differentiate our work from TuNAS, OFA and BigNAS.
- Revised Sec. 4.3. super-net loss landscape to include more context.
- Reran Sec. 4.4. low fidelity experiments to have an equal computational budget. We show that our original conclusions also hold in this setting.
- Add a new baseline for Sec. 4.5, and show that our approach of disabling dynamic channels is still superior.
- Updated Figure 1, 4. and its caption.
- Minor fixes to the text.

---

### Decision · Program_Chairs · 2021-01-07
**Final Decision**

**Decision:**

Reject

**Comment:**

This paper presents an analysis of different tricks for training the super-network in NAS. While all reviewers see value in some of the many experiments, all reviewers also have substantial criticisms of the paper, and all reviewers gave weak rejection scores.

Looking at the paper myself, I agree with this assessment. Several of the experiments are valuable, but there are also several substantial issues.

One question that confused two reviewers and myself is about using sparse Kendall's tau as a metric that the authors in the rebuttal again state can be computed during super-net training to evaluate the quality, just like super-net accuracy. I don't see how that is possible. Kendall's tau measures the correlation between the ranks of the performances of the stand-alone architectures and the ordering implied by the super-net. Computing this requires access to the performance ranks of the stand-alone architectures. For tabular benchmarks this is of course available, but not in practical NAS applications.

I would also like to echo the concern of AnonReviewer2 that too little information is given to fully understand what is shown in Figure 10.

Some reviewers also questioned inhowfar the results generalize to the setting of the Once-for-all-network or BigNAS. This was not a deciding factor for me, since insights based on NAS-Bench101, 201 and a DARTS-like search spaces are already very useful.

I agree with the reviewers that the authors' use of "proxy" is highly misleading. It is standard to refer to the low-fidelity model used for training as the proxy model. In contrast, the authors use it for the final evaluation model.

Concerning the authors' five final take-aways:
1) I don't see how sparse Kendall's tau is actionable.
2) The batch normalization part is interesting, and I agree with the authors that it is useful to spell this out and analyze it, rather than just having one sentence in the paper as NB-201 and TuNAS, but the attribution that this has been done before is broken. "In contrast to X", rather than "Like X"
3) This is interesting, although I agree with AnonReviewer3 that I'm lacking intuition why a smaller learning rate should be useful for a less smooth space
4) The experiment on low fidelity estimates is very misleading. The proxy settings used during training are already low fidelity evaluations -- for the final evaluation, you would increase the number of channels, number of layers and number of epochs. Stating that the use of low fidelities is not useful is highly misleading. The authors' experiments only shows that the proxy model is already well chosen, and that if you reduce #layers or #channels and proportionally increase #epochs, performance gets worse. I encourage the authors to try searching without this proxy model, and I'm sure they will find that (which correlations might increase) the search process will be far too slow.
5) The insight on dynamic channeling appears very useful to me.

In summary, I recommend rejection and encourage the authors to address the points raised by the reviewers and in this meta-review.